# Induction of Senescence by Loss of Gata4 in Cardiac Fibroblasts

**DOI:** 10.3390/cells12121652

**Published:** 2023-06-17

**Authors:** Zhentao Zhang, Gabriella Shayani, Yanping Xu, Ashley Kim, Yurim Hong, Haiyue Feng, Hua Zhu

**Affiliations:** 1Department of Surgery, Davis Heart and Lung Research Institute, The Ohio State University Wexner Medical Center, Columbus, OH 43210, USA; yanping.xu@osumc.edu; 2Division of Cardiovascular Medicine, Department of Medicine, Vanderbilt University Medical Center, Nashville, TN 37232, USA; gabriella.i.shayani@vanderbilt.edu (G.S.); ashley.h.kim@vanderbilt.edu (A.K.); yurim.hong@vanderbilt.edu (Y.H.); haiyue.feng@vumc.org (H.F.)

**Keywords:** senescence, fibroblast, Gata4, heart, Tcf21

## Abstract

Cardiac fibroblasts are a major source of cardiac fibrosis during heart repair processes in various heart diseases. Although it has been shown that cardiac fibroblasts become senescent in response to heart injury, it is unknown how the senescence of cardiac fibroblasts is regulated in vivo. *Gata4,* a cardiogenic transcription factor essential for heart development, is also expressed in cardiac fibroblasts. However, it remains elusive about the role of Gata4 in cardiac fibroblasts. To define the role of Gata4 in cardiac fibroblasts, we generated cardiac fibroblast-specific *Gata4* knockout mice by cross-breeding *Tcf21-MerCreMer* mice with *Gata4^fl/fl^* mice. Using this mouse model, we could genetically ablate *Gata4* in Tcf21 positive cardiac fibroblasts in an inducible manner upon tamoxifen administration. We found that cardiac fibroblast-specific deletion of *Gata4* spontaneously induces senescence in cardiac fibroblasts in vivo and in vitro. We also found that Gata4 expression in both cardiomyocytes and non-myocytes significantly decreases in the aged heart. Interestingly, when *αMHC-MerCreMer* mice were bred with *Gata4^fl/fl^* mice to generate cardiomyocyte-specific Gata4 knockout mice, no senescent cells were detected in the hearts. Taken together, our results demonstrate that *Gata4* deficiency in cardiac fibroblasts activates a program of cellular senescence, suggesting a novel molecular mechanism of cardiac fibroblast senescence.

## 1. Introduction

Cardiac fibroblasts are a major cellular population that regulates extracellular matrix (ECM) production and cardiac fibrosis [1,2]. In response to cardiac injury, cardiac fibroblasts become activated and differentiate into myofibroblasts. The resulting myofibroblasts are the primary source of the extracellular matrix. Excessive extracellular matrix deposition in response to heart injury leads to cardiac fibrosis. Although cardiac fibrosis is initially an adaptive response, its progression disrupts the mechanical architecture of the heart and thus leads to adverse remodeling and heart failure [3,4]. Therefore, extensive studies have focused on the differentiation of cardiac fibroblasts in the context of heart injury. However, much less is known about how the homeostasis of cardiac fibroblasts is regulated in vivo.

Cellular senescence is a programmed cell cycle arrest that prevents the proliferation of old, damaged, and potentially tumorigenic cells [5,6,7]. As such, cellular senescence was initially recognized as a potent tumor-suppressive mechanism that arrests the growth of cells at risk for malignant transformation. However, numerous recent studies identified diverse roles of cellular senescence in the context of physiological or pathological processes beyond tumor suppression [5,6,7,8]. Importantly, cellular senescence has been shown to play important roles in fibrotic wound repair processes in multiple tissues, including the liver [9], skin [10,11], lung [12], heart [13,14,15,16], and spinal cord [17]. It has been shown that activated cardiac fibroblasts or myofibroblasts become senescent in response to heart injuries [13,14,15]. It was demonstrated that senescence of activated cardiac fibroblasts or myofibroblasts is significantly decreased in the global p53 or p53/p16 double knockout mouse model, indicating that senescence of these cardiac fibroblast-derived cells is mediated by p53 and/or p16 pathways [13,14]. These studies also suggested that inhibition of senescence by inactivating p53 or both p16 and p53 increases cardiac fibrosis. In contrast, a recent study showed that eliminating senescent cells using synolytics can decrease scar formation and improve heart function after ischemia-reperfusion injury [16]. Therefore, although the precise roles of cellular senescence during heart injury need to be further investigated, it becomes clear that the senescence of cardiac fibroblast-derived cells (i.e., activated cardiac fibroblasts and myofibroblasts) is involved in heart repair processes. However, it remains elusive about the molecular regulation of senescence of cardiac fibroblasts.

*Gata4*, one of the earliest genes expressed during heart development, plays critical roles in cardiomyocyte proliferation, differentiation, survival, and hypertrophy [18,19,20,21,22,23,24,25,26]. Although Gata4′s roles in the heart have been exclusively studied for cardiomyocytes, it has been shown that *Gata4* mRNA and protein level in isolated cardiac fibroblasts is comparable to or even higher than that in the whole heart of adult mice [27,28]. In contrast, *Gata4* expression was very low in tail-tip or skin fibroblasts, suggesting that Gata4 abundance in fibroblasts is cardiac-specific [27,28]. A recent study has suggested that Gata4, together with Gata6 in periostin lineage traced cells (i.e., activated cardiac fibroblasts and myofibroblasts), contributes to enhancing angiogenesis in a pressure overload mouse model [29]. However, Gata4′s role in cardiac fibroblast homeostasis is still unknown.

In this study, we sought to determine the role of Gata4 in cardiac fibroblasts using a cardiac fibroblast-specific, inducible *Gata4* knockout mouse model. We found that loss of Gata4 in cardiac fibroblasts induces senescence without heart injury. In addition, the role of Gata4 in cellular senescence seems to be cardiac fibroblast specific, as cellular senescence was not observed in the heart of a cardiomyocyte-specific, inducible *Gata4* knockout mouse model. Our results suggest that Gata4 may be an important gatekeeper for the cellular senescence of cardiac fibroblasts. Thus, it might be a potential target to treat cardiac fibrosis.

## 2. Materials and Methods

### 2.1. Mice

All studies conformed with the principles of the National Institutes of Health “Guide for the Care and Use of Laboratory Animals”, (NIH publication No. 85-12, revised 1996). The mice were given unrestricted access to food and water and were housed under a 12 h light/dark cycle at a consistent room temperature of 24 ± 2 °C, with a steady humidity of 40 ± 15%. *Tcf21^iCre^*:*R26R^tdT^* and *Tcf21^iCre^*:*Gata4^fl/fl^* mice were generated by cross-breeding *Tcf21^iCre^* [30,31] with Rosa26-CAG-loxP-Stop-loxP-tdTomato mice (Jackson Laboratory, Bar Harbor, ME, USA) and *Gata4^fl/fl^* mice [32], respectively. Subsequently, *Tcf21^iCre^*:*Gata4^fl/fl^*:*R26R^tdT^* mice were generated by cross-breeding *Tcf21^iCre^*:*R26R^tdT^* mice with *Gata4^fl/fl^* mice. *αMHC^iCre^*:*Gata4^fl/fl^* mice were generated by cross-breeding *αMHC^iCre^* mice (The Jackson Laboratory) with *Gata4^fl/fl^* mice. In this study, 7-week-old and 72-week-old male and female mice were used for the experiments. Tamoxifen (Sigma) was dissolved in corn oil at a concentration of 10 mg/mL by shaking overnight at 37 °C with light protection. To induce Cre activity, tamoxifen was administered to the mice via intraperitoneal injection for 5 consecutive days (75 mg/kg for *Tcf21^iCre^* and 40 mg/kg for *αMHC^iCre^*). 8-weeks and 72-week-old C57BL6 mice were obtained from the Jackson Laboratory. Carbon dioxide (CO_2_) inhalation was used as a primary euthanasia, followed by cervical dislocation as a secondary measure.

### 2.2. Immunohistochemistry

After the hearts were washed with ice-cold PBS, they were fixed with pre-chilled 4% paraformaldehyde (PFA) in PBS overnight at 4 °C and then incubated in 30% sucrose in PBS on a rotary shaker overnight at 4 °C. The fixed hearts were embedded in O.C.T compound and frozen in pre-chilled isopentane with liquid nitrogen. The frozen hearts were sectioned at 8 µm thickness. After 10 min of air drying, the heart sections were washed with PBS three times and then fixed again with pre-chilled 4% PFA in PBS on ice for 20 min. After PBS washes thrice, the fixed heart sections were permeabilized in 0.1% Triton X-100 in PBS for 20 min. The heart sections were washed three times with PBS and then blocked with M.O.M mouse IgG blocking reagent (Vector Labs, Newark, CA, USA) for 1 h and 5% donkey serum in M.O.M protein diluent (Vector Labs) for 30 min at room temperature. The heart sections were incubated with anti-Troponin I (Abcam, Waltham, MA, USA, cat# ab188877, 1:400 dilution), anti-Gata4 (Thermofisher, Waltham, MA, USA, cat# PA1-102, 1:200 dilution), anti-p16 (Santa Cruz, Dallas, TX, USA, cat# sc56330, 1:200 dilution), or indicated combinations of the primary antibodies for 1 h at room temperature. After the heart sections were washed with PBS three times, they were incubated with AlexaFluor secondary antibodies (Thermofisher) for one hour at room temperature and mounted with Vectashield antifade mounting medium with DAPI (VectorLabs, Newark, CA, USA, cat# H-2000). tdTomato was visualized without immunostaining. For wheat germ agglutinin (WGA) staining, the heart sections were incubated with WGA staining solution (Thermofisher, cat# W32464) prepared per manufacturer’s instruction, along with secondary antibodies. Images were captured using Zeiss LSM 880 confocal microscope or Olympus IX81 epifluorescent microscope. A total of 6 heart sections per mouse were used for quantification. Fluorescent pictures of four randomly selected fields per section were captured and analyzed by MetaExpress software (Molecular Device, San Jose, CA, USA) in an automated manner.

### 2.3. Senescence-Associated β-Galactosidase (SA β-Gal) Staining

Cryo-embedded heart sections were prepared as described above and stained using a SA β-gal staining kit (Sigma, Setagaya city, Japan, cat# CS0030-1KT) per the manufacturer’s protocol. Briefly, frozen heart sections were fixed with 4% PFA for 10 min at room temperature and washed thrice with PBS. The heart sections were incubated with freshly prepared SA β-gal staining solution (per the kit’s instruction) overnight at 37 °C. After stained heart sections were washed with PBS thrice, the pictures were captured using Leica DMIL LED, *n* = 5.

### 2.4. Isolation of Non-Myocytes Using Langendorff Perfusion

Adult mouse hearts were dissected and perfused retrogradely via aortic cannulation in a Langendorff apparatus as described previously [33]. The dissected hearts were serially perfused with perfusion buffer (NaCl 120.4 mM, KCl 14.7 mM, Na_2_HPO_4_ 0.6 mM, Ka_2_HPO_4_ 0.6 mM, MgSO_4_ 1.2 mM, Na-HEPES 10 mM, NaHCO_3_ 4.6 mM, Taurine 30 mM, BDM 10 mM, Glucose 5.5 mM, pH 7.0), digestion buffer without CaCl_2_ (Collagenase II 2.4 mg/mL in perfusion buffer), and digestion buffer with CaCl_2_ (Collagenase II 2.4 mg/mL, and CaCl_2_ 40 µM in perfusion buffer). The perfused hearts were displaced from the Langendorff apparatus. Ventricular cardiomyocytes were mechanically dissociated and triturated using a fine scalpel and scissors and resuspended in stopping buffer (CaCl_2_ 11.7 µM in calf serum 2 mL plus perfusion buffer 18 mL). Cells were centrifuged at a low speed, and the supernatant was collected as a non-myocyte population. The supernatant was centrifuged at 350× *g* for 10 min to obtain non-myocytes, *n* = 3.

### 2.5. Neonatal Cardiac Fibroblast Isolation

Neonatal cardiac fibroblasts were isolated using the Neomyts kit (Cellutron, Baltimore, MD, USA, cat# nc-6031). Following the digestion of the hearts per the manufacturer’s protocol, we harvested cells containing a mix of cardiomyocytes and cardiac fibroblasts. These mixed cells were then plated in a 6-well plate and allowed to settle for 40 min, enabling the cardiac fibroblasts to attach to the bottom. The supernatant, which contains cardiomyocytes, was discarded. The cells were subsequently cultured in Dulbecco’s Modified Eagle’s Medium (DMEM) (Gibco, Grand Island, NY, USA, cat# 10564011), supplemented with 10% fetal bovine serum (FBS) (Corning, NewYork, NY, USA, cat# 35-070-CV) and 1% penicillin-streptomycin (Sigma, P0781).

### 2.6. Adenovirus Infection

Non-myocytes isolated using Langendorff perfusion described above were plated into a 10 cm culture dish. Mostly cardiac fibroblasts were attached to the culture dish. Cardiac fibroblasts were cultured until they reached ~70–80% confluency. Then, cardiac fibroblasts were trypsinized, replated into a 24-well plate, and then infected with Ad-vector (Ad5CMVempty, VVC-U of Iowa-272) or Ad-Cre (Ad5-CMV-Cre, VVC-U of Iowa-5) viruses purchased from the University of Iowa Viral Core following the core’s Adenovirus infection protocol (www.medicine.uiowa.edu/vectorcore (accessed on 12 November 2020)).

### 2.7. Immunocytochemistry

After cells were fixed with 4% paraformaldehyde for 15 min, they were washed with permeabilization buffer (0.05% Triton-X in PBS) for 5 min three times at room temperature. The fixed cells were incubated with blocking buffer (BiogeneX, Fremont, CA, USA, cat# HK085) for 45 min, and then incubated with primary antibodies against Gata4 (Thermo Scientific, Waltham, MA, USA, cat# PA1-102, 1:400 dilution), Ki67 (Abcam, cat# ab92742, 1:1000 dilution), αSMA (Sigma, cat# A5228, 1:1000 dilution), or γH2AX (Novus Biologicals, Centennial, CO, USA, cat#NB100-384, 1:200) overnight at 4 °C. After being washed with permeabilization buffer three times for 5 min, cells were incubated with Alexa fluorogenic secondary antibodies (Thermofisher) at 1:400 dilution at room temperature for 1 h. For each animal, five slides ranging from the top to the bottom of the heart were selected for staining. Cells were rewashed with permeabilization buffer three times and then incubated with DAPI solution at a final concentration of 2 µM in permeabilization buffer for 15 min. After cells were washed three times with permeabilization buffer, cell images were captured with ImageXpress Micro XL Automated Cell Imaging system (Molecular Device), *n* = 3.

### 2.8. EdU Incorporation Assay

EdU incorporation assay was performed using Click-iT Plus EdU imaging kit (Thermo Fisher, cat# C10637) per the manufacturer’s protocol. Briefly, cardiac fibroblasts were cultured in a fibroblast growth medium supplemented with 10 µM EdU for 24 h. After cells were fixed with 4% paraformaldehyde for 15 min, they were washed with 3% BSA in PBS twice at room temperature. The fixed cells were permeabilized with 0.5% Triton-X in PBS for 20 min at room temperature and then incubated with freshly prepared EdU staining solution (as per kit’s instruction) for 30 min with light protection, *n* = 3.

### 2.9. High Content Imaging and Analysis

High-content imaging analysis was performed as described previously [34,35,36,37]. The images of immunostained cells were acquired with a 10× objective at 36 fields per well using ImageXpress Micro XL Automated Cell Imaging system (Molecular Device) using DAPI and Texas Red or FITC filter sets. The number of Texas Red or FITC fluorescent positive cells among DAPI fluorescent positive cells was quantified using MetaXpress software (Molecular Device), *n* = 5.

### 2.10. Echocardiography

Echocardiographic imaging was performed by skilled technicians at the Vanderbilt Cardiovascular Physiology Core using the Visual Sonics Vevo 2100 small animal imaging system. Mice were lightly anesthetized using 1.5% isoflurane and placed supine on a heating pad. Parasternal short-axis M-mode imaging was used to visualize LV structure and function for the heart cycle.

### 2.11. FACS

Non-myocytes were isolated from *Tcf21^iCre^:Gata4^fl/fl^:R26RtdT*, *Tcf21^iCre^:R26R^tdT^*, or wild-type mice using Langendorff perfusion as described above. tdTomato^+^ cells were sorted on FACSaria III (BD Biosciences). The gate for tdTomato^+^ cells was set using tdTomato- cells isolated from wild-type mice. Sorted tdTomato^+^ cells were used for qPCRs to determine the mRNA expression level of *Gata4*, *n* = 3.

### 2.12. Quantitative Real-Time PCR (qPCR)

Total RNA was isolated from sorted tdTomato^+^ cells using NucleoSpin RNA Kit (Macherey-Nagel). cDNA was synthesized by reverse transcription qPCR using High-Capacity cDNA Reverse Transcription Kit (Applied Biosystems, Waltham, MA, USA, cat# 4368814). qPCR analyses were performed using SYBR probes and iTaq Universal SYBR Green Supermix (Bio-Rad, Hercules, CA, USA, cat# 1725121) on a Bio-Rad CFX96 system (Bio-Rad). The gene expression level was normalized to GAPDH.

### 2.13. Statistical Analyses

Statistical significance was determined using unpaired two-tailed Student’s t-test between two groups or one-way ANOVA with Tukey’s post hoc analysis among three groups. *p*-values of < 0.05 were regarded as significant. GraphPad Prism 8 was used for data visualization and statistical analysis.

## 3. Results

In previous studies, Gata4 is known to be expressed in cardiac fibroblasts [27,28]. Its expression level in cardiac fibroblasts was much higher than the fibroblasts derived from other organs and surprisingly even higher than that in the whole adult mouse heart [27]. Given the heterogeneous nature of cardiac fibroblasts, we sought to precisely determine Gata4 expression in cardiac fibroblasts using cardiac fibroblast lineage reporter mice that were generated by cross-breeding *Tcf21-MerCreMer* mice with *Rosa26-CAG-loxP-Stop-loxP-tdTomato* mice (referred to as *Tcf21^iCre^*:*R26R^tdT^* mice). Tcf21, an epicardial transcription factor, is expressed in epicardium-derived cardiac fibroblasts, constituting more than 80% of cardiac fibroblasts in the heart [38,39]. In *Tcf21^iCre^*:*R26R^tdT^* mice, cardiac fibroblasts are specifically labeled with tdTomato upon tamoxifen administration [30]. A week after tamoxifen administration, we processed heart sections and performed immunostaining for Gata4. We identified Gata4^+^tdTomato^+^ Tcf21 lineage traced cells (Figure 1A). Our results showed that Gata4 indeed expresses in Tcf21 lineage traced cardiac fibroblasts.

To investigate the role of Gata4 in cardiac fibroblasts, we generated inducible, cardiac fibroblast-specific Gata4 knockout mice by cross-breeding *Tcf21-MerCreMer* mice with *Gata4^fl/fl^* mice (referred to as *Tcf21^iCre^*:*Gata4^fl/fl^* mice) (Figure 1B). We also generated *Tcf21^iCre^*:*Gata4^fl/fl^*:*R26R^tdT^* mice by cross-breeding *Tcf21^iCre^*:*R26R^tdT^* mice with *Gata4^fl/fl^* mice to identify Gata4 deficient cardiac fibroblasts with tdTomato expression. This conditional knockout approach bypasses the embryonic lethality caused by germline Gata4 deficiency [21,24] and thus allows us to study the effect of Gata4 loss in adult cardiac fibroblasts. Using these mouse models, we could genetically ablate *Gata4* in an inducible manner upon tamoxifen administration. A week after administering tamoxifen into *Tcf21^iCre^*:*R26R^tdT^* (control) and *Tcf21^iCre^*:*Gata4^fl/fl^*:*R26R^tdT^* mice for 5 consecutive days, we processed frozen heart sections and immunostained for Gata4. While Gata4 was frequently expressed in tdTomato^+^ cells in Tcf21iCre:R26RtdT mice, tdTomato^+^ cells rarely expressed Gata4 in *Tcf21^iCre^*:*Gata4^fl/fl^*:*R26R^tdT^* mice (Figure 1C,D). We also quantified the extent of genetic deletion of Gata4 in Tcf21 lineage-traced cells using qPCR (Figure 1E). After isolating non-myocytes from *Tcf21^iCre^*:*R26R^tdT^* (control) and *Tcf21^iCre^*:*Gata4^fl/fl^*:*R26R^tdT^* mice, we sorted out tdTomato^+^ cells using fluorescence-activated cell sorting (FACS) (Appendix A). Using the sorted tdTomato^+^ cells from control and *Tcf21^iCre^*:*Gata4^fl/fl^*:*R26R^tdT^* mice, we performed qPCR to assess the *Gata4* mRNA level. We found ~80% reduction of *Gata4* mRNA level in *Tcf21^iCre^*:*Gata4^fl/fl^*:*R26R^tdT^* mice compared to that in *Tcf21^iCre^*:*R26R^tdT^* mice (Figure 1E). Taken together, our results demonstrated that our conditional *Gata4* knockout model efficiently inactivates Gata4, specifically in cardiac fibroblasts in vivo.

Given Gata4′s positive regulation of cell cycle progression in highly proliferating, multiple different cell types [40,41,42,43,44,45,46,47], we tested if cardiac fibroblast-specific deletion of Gata4 can induce senescence in quiescent cardiac fibroblasts in the heart. We administered tamoxifen to *Tcf21^iCre^* (control) and *Tcf21^iCre^*:*Gata4^fl/fl^* mice for 5 consecutive days. Two or seven weeks after the completion of tamoxifen administration, we processed frozen heart sections and stained them for senescence-associated β-gal (SA β-gal) to evaluate cellular senescence. Surprisingly, we found a significant number of SA β-gal+ interstitial cells in *Tcf21i^Cre^*:*Gata4^fl/fl^* mouse hearts, while nearly no senescent cells were observed in control hearts (Figure 2A,C). We also confirmed this result by immunostaining for p16, a widely used senescent marker [48]. P16, another senescent marker, was consistently markedly increased in *Tcf21^iCre^*:*Gata4^fl/fl^* mouse hearts compared to control hearts (Figure 2B,D). Lack of senescence in *Tcf21^iCre^* control hearts excludes the possibility that Cre toxicity could induce senescence. Our results showed that specific deletion of Gata4 in cardiac fibroblasts spontaneously induces cellular senescence in the heart in vivo. These findings suggest that Gata4 may play a role in the regulation of cellular senescence in quiescent cardiac fibroblasts, while it contributes to cell proliferation in highly proliferative cells, including embryonic cardiomyocytes and intestinal epithelial cells and cancer cells [40,41,44,45,46,47].

To determine whether senescence induced by cardiac fibroblast-specific Gata4 loss in the heart could be observed at the cellular level in a cell-autonomous manner, we also tested if loss of Gata4 in isolated primary cardiac fibroblasts can result in senescence induction in vitro. We isolated adult cardiac fibroblasts from *Gata4^fl/fl^* mice. To genetically delete Gata4 in isolated cardiac fibroblasts, isolated *Gata4^fl/fl^* cardiac fibroblasts were infected with a Cre-expressing adenovirus (Ad-Cre) or control adenovirus (Ad-vector). Three days after the adenoviral infection, we showed efficient Gata4 inactivation in Ad-Cre infected cells (Figure 3A). We rarely observed Gata4 expressing cells with Ad-Cre infection, while most cells expressed Gata4 with Ad-vector infection. Using this cell-based *Gata4* knockout model, we first examined cell proliferation using EdU labeling and Ki67 immunostaining. Since senescent cells are withdrawn from the cell cycle, cellular senescence inversely correlates with cell proliferation rate. We found that both EdU^+^ and Ki67^+^ cells were significantly decreased by Ad-Cre infection, compared with Ad-vector infection, indicating that cell proliferation rate is reduced by the loss of Gata4 in isolated cardiac fibroblasts (Figure 3B). Next, we performed SA β-gal staining 10 days after infecting Ad-Cre or Ad-vector into *Gata4^fl/fl^* cardiac fibroblasts. We observed markedly increased SA β-gal^+^ cells by Ad-Cre infection, compared to Ad-vector infection (Figure 3C). As parallel experiments, we performed SA β-gal staining on wild-type cardiac fibroblasts infected with Ad-Cre. In contrast to *Gata4^fl/fl^* cardiac fibroblasts infected with Ad-Cre, these cells did not show an appreciable increase in the percentage of senescent cells over the background level. These results indicate that senescence induction is indeed due to the loss of Gata4 rather than a non-specific Cre toxicity. In addition, we immunostained the cells infected with Ad-Cre or Ad-vector for γH2AX, a senescence marker. As expected, we found that significantly increased γH2AX^+^ cells in Gata4 deficient cells (Ad-Cre infected) than control cells (Ad-vector infected) (Figure 3C). We also use p16, another senescence marker, for the staining (Appendix A). Taken together, our results demonstrated that loss of Gata4 induces senescence in isolated cardiac fibroblasts in vitro, suggesting that induction of cellular senescence by Gata4 loss is a cell-autonomous effect.

We examined the temporal expression pattern of Gata4 in the heart using mice of different ages. Interestingly, we found that the number of Gata4-expressing cells, including both cardiomyocytes and non-myocytes, is significantly decreased in the old mouse hearts (Appendix A). While Gata4 expression was robust in the neonatal heart, it was sparsely expressed in the 72 weeks old mouse heart. To determine if Gata4 in cardiomyocytes also plays a role in the repression of senescence, we generated cardiomyocyte-specific, inducible *Gata4* knockout mice by cross-breeding *αMHC-MerCreMer* mice [49] with *Gata4^fl/fl^* mice (referred to as *αMHC^iCre^*:*Gata4^fl/fl^*) (Figure 4A). Following tamoxifen administration, we could genetically ablate Gata4, specifically in cardiomyocytes. Nearly no Gata4 expressing cardiomyocyte was observed in *αMHC^iCre^*:*Gata4^fl/fl^* mice, while Gata4 was frequently expressed in cardiomyocytes of *αMHC^iCre^* (control) mice (Figure 4B). However, Gata4^+^ interstitial cells were similarly found in both *αMHC^iCre^*:*Gata4^fl/fl^* and *αMHC^iCre^* (control) mice. We found that the total number of Gata4^+^ cells is significantly reduced in *αMHC^iCre^*:*Gata4^fl/fl^* mouse hearts as opposed to *αMHC^iCre^* (control) mouse hearts (Figure 4C and Appendix A). Collectively, our data demonstrated that we could inactivate Gata4 specifically in cardiomyocytes in an inducible manner using the *αMHC^iCre^*:*Gata4^fl/fl^* mouse model. Next, we examined senescence induction using SA β-gal staining and p16 immunostaining 7 weeks after tamoxifen administration into *αMHC^iCre^*:*Gata4^fl/fl^* and αMHCiCre mice. In contrast to cardiac fibroblast-specific *Gata4* knockout mice (Figure 2), we observed nearly no senescent cells in *αMHC^iCre^*:*Gata4^fl/fl^* mouse hearts in which Gata4 was specifically lost in cardiomyocytes (Figure 4D,E). The transthoracic echocardiogram results reveal no reduction in contractile function in the hearts of cardiomyocyte-specific *Gata4* knockout mice (Appendix A). These results suggest that Gata4′s role in senescence inhibition is specific to cardiac fibroblasts.

## 4. Discussion

In this study, we demonstrated that genetic disruption of *Gata4* in cardiac fibroblasts spontaneously induces cellular senescence. This result indicates that Gata4 deficiency in cardiac fibroblasts activates a program of cellular senescence, suggesting a Gata4′s senescence-suppressive role in cardiac fibroblasts. However, it seems that Gata4′s inhibitory function in cellular senescence is cardiac fibroblast specific, given that we did not observe cellular senescence in a cardiomyocyte-specific Gata4 deficient mouse model. While the mechanisms underlying senescence induction in response to various cellular stresses have been extensively studied, our understanding of how cellular senescence is controlled in healthy tissues is very limited. Our results provide initial evidence that there is a counter-regulatory mechanism of senescence of cardiac fibroblasts in homeostasis. Gata4 expression may be necessary to prevent premature cellular senescence in young and healthy cardiac fibroblasts. Markedly decreased Gata4 expression may prime old cardiac fibroblasts to activate a senescence program upon cellular stresses.

A limitation of our study is that we did not identify the molecular mechanisms underlying senescence induction by loss of Gata4. Since Cdk4 and Cyclin D2 are direct transcriptional targets of Gata4 [40,45,50], it can be speculated that these cell cycle proteins may be involved in cardiac fibroblast senescence induced by Gata4 loss (Appendix A). Understanding how Gata4 represses the senescence of cardiac fibroblasts may allow us to develop anti-senescence strategies in the future. In addition, the functional significance of senescent cardiac fibroblasts remains elusive. Although we showed an inverse correlation between Gata4 expression level and the age of mice, our study did not reveal the functional significance of cardiac fibroblast senescence in the uninjured heart. Since cellular senescence has been shown to have both beneficial and detrimental effects [8], it would be interesting to investigate the functional consequence of senescent cardiac fibroblasts in homeostasis.

Kang et al. demonstrated that Gata4 overexpression spontaneously induces senescence, while *Gata4* knockdown decreases cellular senescence induced by ionizing radiation in IMR-90 human fetal lung and BJ human foreskin fibroblast lines in vitro [51]. Although our results may be in clear contrast with the report from Kang et al., there were a couple of important differences between the previous study and ours. First, the Gata4 expression level in cardiac fibroblasts is much higher than those derived from other organs [27,28]. It is conceivable that Gata4 may play differential roles depending on the type of tissues. For example, while Gata4 in cardiomyocytes is essential for heart development, hypertrophy, and cell survival, cardiomyocyte-specific loss of Gata4 does not induce senescence. Second, the proliferative capacity of the cells used in our study is much lower than that of human fibroblast lines used in the previous study [51]. Population doubling of IMR-90 and BJ cell lines is about 70–80 times [51,52,53], while cardiac fibroblasts in the heart are quiescent without injury, and the population doubling of isolated primary cardiac fibroblasts is less than five times in our hands. Collectively, we speculate that Gata4 may regulate the balance between cell cycle progression and arrest (senescence) depending on proliferative capacity, senescence susceptibility, and/or its expression level in cells.

## 5. Conclusions

In this work, we defined the role of Gata4 in cardiac fibroblasts. By using cardiac fibroblast specific *Gata4* knockout mice, we demonstrated that *Gata4* deficiency in cardiac fibroblasts activates a program of cellular senescence. This suggests a novel molecular mechanism of cardiac fibroblast senescence.

## Figures and Tables

**Figure 1 cells-12-01652-f001:**
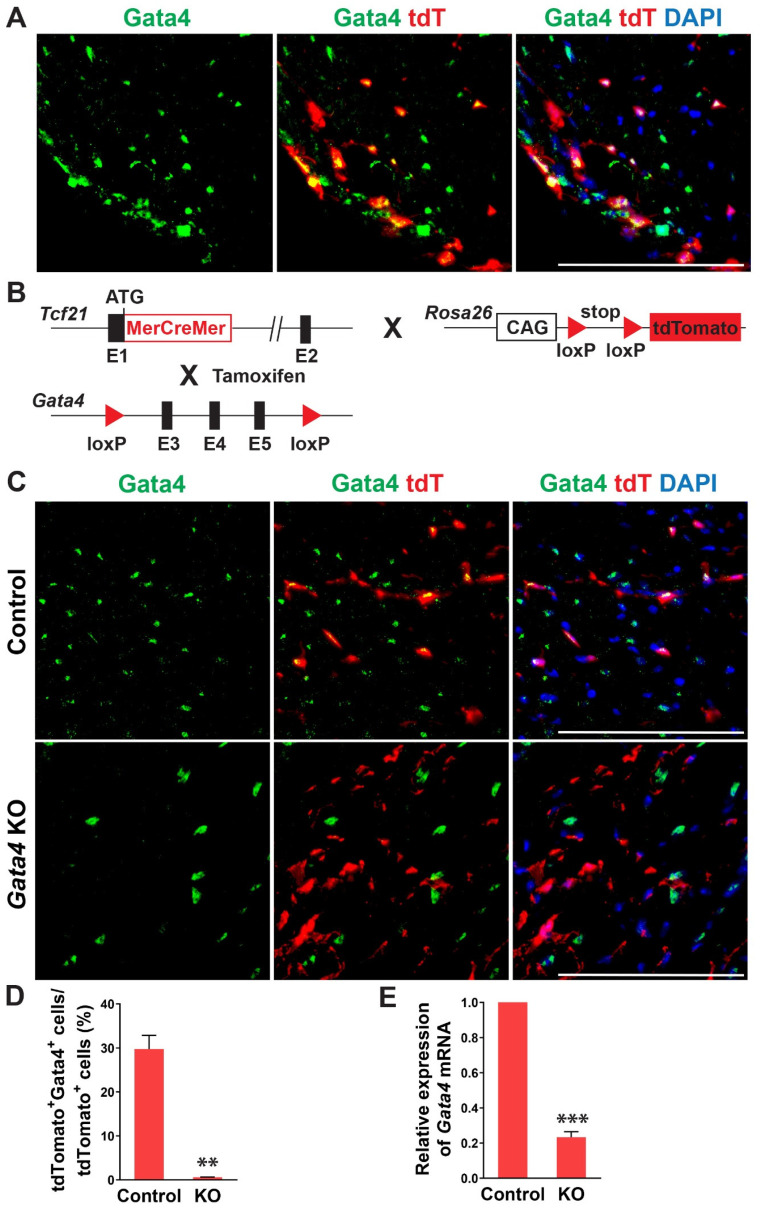
Gata4 expression and inactivation in Tcf21 lineage traced cardiac fibroblasts. (**A**) Gata4 expression in Tcf21 lineage traced cardiac fibroblasts. Following tamoxifen administration, 8-weeks old *Tcf21^iCre^*:*R26R^tdT^* mouse hearts were harvested, and heart sections were processed and immunostained for Gata4. Scale bar, 100 µM. (**B**) Schematic illustration of generation of cardiac fibroblast specific, inducible *Gata4* knockout mice. *Tcf21^iCre^* mice in which *MerCreMer* cDNA cassette was inserted into exon 1 (E1) were crossed with Gata4 floxed mice which contain a pair of loxP sites encompassing exons 3, 4, and 5 (E3, 4, and 5) of Gata4. Alternatively, *Tcf21^iCre^*:*R26R^tdT^* mice were crossed with Gata4 floxed mice. Activated *MerCreMer* by tamoxifen administration recombines two loxP sites to delete exons 3, 4, and 5 of *Gata4*, thereby inactivating Gata4 in Tcf21 lineage traced cardiac fibroblasts. (**C**) Immunofluorescent analysis to demonstrate Gata4 inactivation in Tcf21 lineage traced cardiac fibroblasts. Following tamoxifen administration, *Tcf21^iCre^*:*R26R^tdT^* (Control) and *Tcf21^iCre^*:*Gata4^fl/fl^*: *R26R^tdT^* (Gata4 KO) mouse hearts were harvested, and heart sections were processed and immunostained for Gata4. Scale bar, 100 µM. (**D**) Quantification of Gata4 expressing cardiac fibroblasts using immunostained heart sections from *Tcf21^iCre^*:*R26R^tdT^* (Control) and *Tcf21^iCre^*:*Gata4^fl/fl^*:*R26R^tdT^* (KO) mice. The percentage of Gata4^+^tdTomato^+^ cells among tdTomato^+^ cells (Tcf21 lineage traced cardiac fibroblasts) was quantified. Three independent experiments are presented as mean ± s.d. *n* = 3, ** *p* < 0.01. (**E**) qPCR analysis for *Gata4* mRNA level using sorted tdTomato^+^ cells from *Tcf21^iCre^*:*Gata4^fl/fl^*:*R26R^tdT^* (KO). FACS-sorted tdTomato^+^ cells from *Tcf21^iCre^*:*R26R^tdT^* (Control) mice were used as control. Three independent experiments are presented as mean ± s.d. *n* = 3, *** *p* < 0.0001.

**Figure 2 cells-12-01652-f002:**
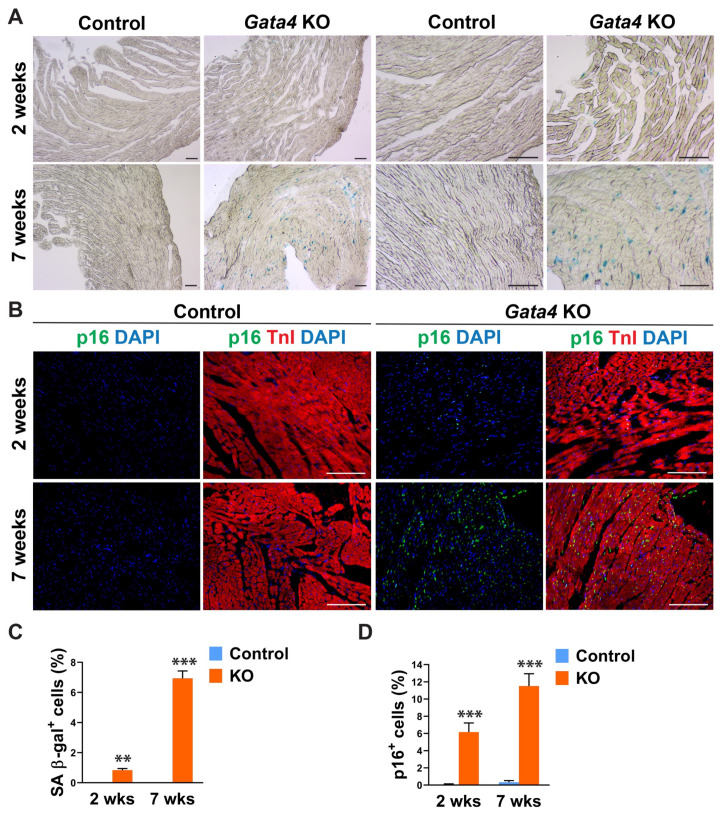
Senescence induction by Gata4 loss in cardiac fibroblasts in the heart. (**A**) Demonstration of senescence by SA β-gal staining. SA β-gal staining was performed on heart sections 2- or 7 weeks following tamoxifen administration into *Tcf21^iCre^*:*Gata4^fl/fl^* (Gata4 KO) or *Tcf21^iCre^* (Control) mice. Scale bar, 40 µM, and 100 µM. (**B**) Demonstration of senescence by p16 immunostaining. The heart sections from *Tcf21^iCre^*:*Gata4^fl/fl^* (*Gata4* KO) and *Tcf21^iCre^* (Control) mice were immunostained for p16 2-week (2 wks) or 7-week (7 wks) following tamoxifen administration. Scale bar, 100 µM. (**C**,**D**) Summary of quantification of senescent cells in cardiac fibroblast specific Gata4 deficient hearts. Three independent experiments are presented as mean ± s.d. *n* = 3, ** *p* < 0.01; *** *p* < 0.001.

**Figure 3 cells-12-01652-f003:**
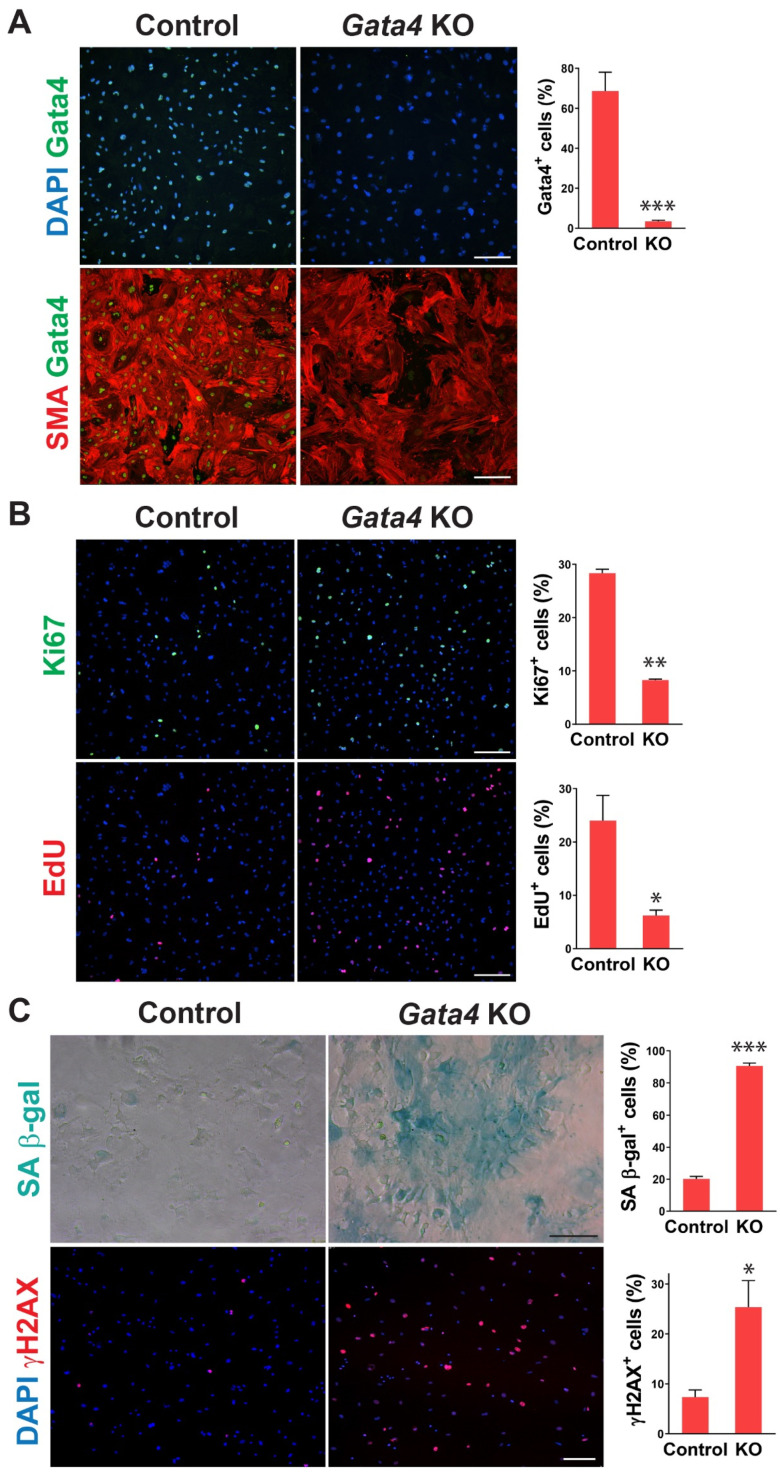
Senescence induction in Gata4 deficient cardiac fibroblasts in vitro. (**A**) Inactivation of Gata4 in vitro. Ad-Cre or Ad-vector viruses were infected into adult cardiac fibroblasts isolated from *Gata4^fl/fl^* mice. The infected cells were immunostained for alpha-smooth muscle actin (SMA) and Gata4. Representative immunofluorescent images of cells infected with Ad-vector (Control) or Ad-Cre (*Gata4* KO) were shown. Gata4-expressing cells were quantified using high-content imaging analysis. Five independent experiments are presented as mean ± s.d. *n* = 5, *** *p* < 0.0001. Scale bar, 200 µM. (**B**) The proliferation of Gata4 deficient cardiac fibroblasts. *Gata4^fl/fl^* cardiac fibroblasts infected with Ad-vector (Control) or Ad-Cre (*Gata4* KO) were EdU labeled or immunostained for Ki67. Representative immunofluorescent images used for high-content imaging analysis were shown. Three independent experiments are presented as mean ± s.d. *n* = 3, * *p* < 0.05; ** *p* < 0.005. Scale bar, 200 µM. (**C**) Demonstration of senescence in Gata4 deficient cardiac fibroblasts. *Gata4^fl/fl^* cardiac fibroblasts infected with Ad-vector (Control) or Ad-Cre (*Gata4* KO) were stained for SA β-gal or immunostained for γH2AX. Representative SA β-gal stained or immunofluorescent images were shown. Five (for SA β-gal) and three (for γH2AX) independent experiments are presented as mean ± s.d. *n* = 5, * *p* < 0.05; *n* = 3, *** *p* < 0.0001. Scale bar, 100 µM.

**Figure 4 cells-12-01652-f004:**
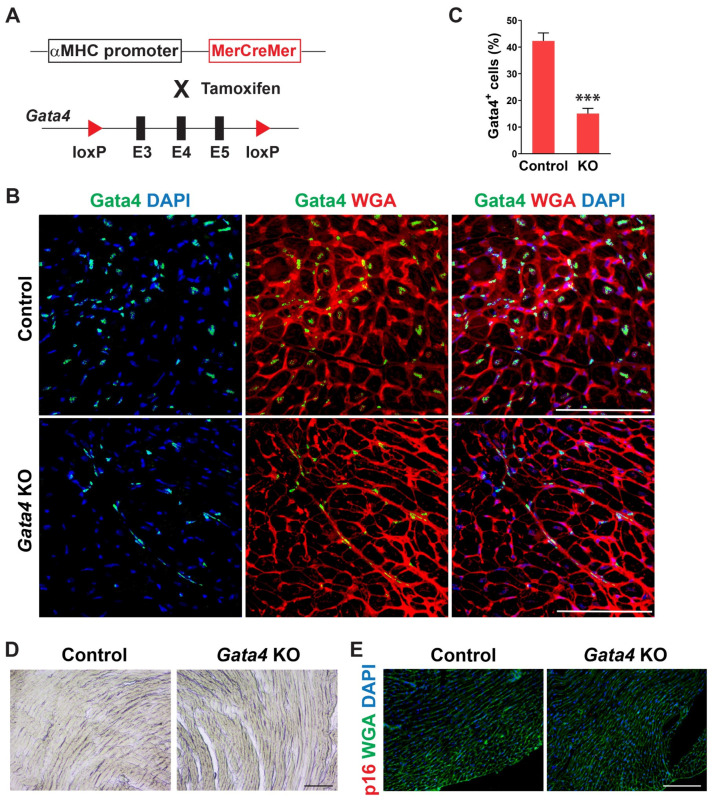
Generation of cardiomyocyte-specific, inducible Gata4 KO mice and evaluation of senescence induction. (**A**) Schematic illustration of generation of cardiomyocyte-specific, inducible *Gata4* knockout mice. The transgenic mice which containing *αMHC* promoter driven *MerCreMer* cDNA cassette and were cross-bred with Gata4 floxed mice which contained a pair of loxP sites encompassing exons 3, 4, and 5 (E3, 4, and 5). Activated *MerCreMer* by tamoxifen administration recombines two loxP sites to delete exons 3, 4, and 5 of *Gata4*, thereby inactivating Gata4 in *αMHC* lineage traced cardiomyocytes. (**B**) Immunofluorescent analysis to demonstrate Gata4 inactivation in *αMHC* lineage traced cardiomyocytes. A week after tamoxifen administration, *αMHC^iCre^* (Control) and *αMHC^iCre^*:*Gata4^fl/fl^* (*Gata4* KO) mouse hearts were harvested, and heart sections were processed. The heart sections were immunostained for Gata4 and stained for wheat germ agglutinin (WGA) to demarcate the cell boundary. Gata4^+^ nuclei in the WGA-stained interstitial area indicate Gata4^+^ interstitial cells. Scale bar, 100 µM. (**C**) Quantification of Gata4^+^ cells in the heart. Gata4^+^DAPI^+^ cells were quantified as Gata4+ cells in the heart. Three independent experiments are presented as mean ± s.d. *n* = 3, *** *p* < 0.0001. (**D**,**E**) No senescence induction in cardiomyocyte-specific Gata4 deficient hearts. SA β-gal staining (**D**) and p16 immunostaining (**E**) were performed on heart sections 7 weeks following tamoxifen administration into *αMHC^iCre^*:*Gata4^fl/fl^* (*Gata4* KO) and *αMHC^iCre^* (Control) mice. Three mice per group were evaluated. Scale bar, 100 µM.

## Data Availability

The original contributions presented in this study are included in the article. Further inquiries can be directed to the corresponding author.

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
