# Peer review of "Induction of Senescence by Loss of Gata4 in Cardiac Fibroblasts"

_cells, 2023, doi:10.3390/cells12121652_

Round 1

Reviewer 1 Report

In this paper entitled, “Induction of Senescence by Loss of Gata4 in Cardiac Fibroblasts” Zhang et al. investigate the role of GATA4, a cardiogenic transcription factor, in cardiac cells using a mouse model to deplete GATA4 upon tamoxifen administration (Fig.1). Their results show that deletion of GATA4 induces senescence in cardiac fibroblasts (Fig.2) in vivo and in vitro (Fig.3), but no senescence was found following loss of GATA4 in cardiomyocytes (Fig.4).

Major comments:

1.) As traditional deletion of Gata4 causes embryonic lethality, is there a connection between GATA4 cardiac fibroblast specific deletion and mouse mortality?

2.) Since Cdk4 and Cyclin D2 are direct transcriptional targets of Gata4, the authors should check if these two genes might be involved in cardiac fibroblast senescence induced by Gata4 loss.  

Minor comments:

3.) Please indicate the n number for every experiment.

4.) Please specify mouse age and gender in the methods.

5.) Please show data for FACS sorting.

English language will be acceptable after final review.

Author Response

In this paper entitled, “Induction of Senescence by Loss of Gata4 in Cardiac Fibroblasts” Zhang et al. investigate the role of GATA4, a cardiogenic transcription factor, in cardiac cells using a mouse model to deplete GATA4 upon tamoxifen administration (Fig.1). Their results show that deletion of GATA4 induces senescence in cardiac fibroblasts (Fig.2) in vivo and in vitro (Fig.3), but no senescence was found following loss of GATA4 in cardiomyocytes (Fig.4).

Major comments:

  • As traditional deletion of Gata4 causes embryonic lethality, is there a connection between GATA4 cardiac fibroblast specific deletion and mouse mortality?

Thank you for your insightful question. Previous studies have shown that whole body Gata4 deletion leads to early embryonic lethality due to defects in heart development and function. Therefore, our study employed the inducible Cre/loxP estrogen receptor (ER) transgenic system, allowing us to specifically delete Gata4 in adult mouse hearts. Throughout our experiment, we did not observe any fatalities resulting from both cardiac fibroblast specific and cardiomyocyte specific deletion of Gata4.

  • Since Cdk4 and Cyclin D2 are direct transcriptional targets of Gata4, the authors should check if these two genes might be involved in cardiac fibroblast senescence induced by Gata4 loss.  

Thank you for your suggestion. We have evaluated the protein and mRNA expression levels of Cdk4 and Cyclin D2 using two distinct genetic models. Firstly, we carried out a genetic knockout of Gata4 by infecting cardiac fibroblasts isolated from Gata4fl/fl mice with Ad-Cre viruses. Then, we conducted immunostaining for Cdk4 and Cyclin D2. Through high content imaging analysis, we were able to quantify Cdk4 or Cyclin D2+ cells. Our findings indicated that, as compared to control cells, a significant decrease of Cdk4 protein was observed in Gata4-/- cardiac fibroblasts, while there was a non-significant trend towards the reduction of Cyclin D2 in Gata4-/- cells (Figure S6A). Furthermore, we analyzed mRNA expression of Cdk4 and Ccnd2 in FACS sorted tdTomato+ cells (Gata4-/- cardiac fibroblasts). Both Cdk4 and Ccnd2 mRNA levels significantly decreased in Gata4-/- cells (Figure S6B). For further details, please refer to Supplementary Materials Figure S6.

Minor comments:

  • Please indicate the n number for every experiment.

Thank you for your suggestion. I have marked the 'n' number for all experiments. Please refer to lines 115, 127, 148, 156, 162, 168, 238, 241, 266, 299, 303, 308, 345, 400, 437, 446, and 449 for this information.

  • Please specify mouse age and gender in the methods.

Thank you for pointing this out. I have now added this information to the Methods section. Please refer to page 2, line 80, for these details.

  • Please show data for FACS sorting.

Thanks for pointing it out. I added FACS results in the supplementary. For further details, please refer to Supplementary Materials Figure S2.

Reviewer 2 Report

In this manuscript the authors explore the role of the cardiogenic transcription factor Gata4 in fibroblast senescence in homeostasis. The authors discovered that Gata4 is responsible for cardiac fibroblast senescence; moreover, this was specific to cardiac fibroblasts and did not occur in cardiomyocytes. These findings are significant and present a new avenue of research for the treatment of fibrosis in the heart.

The method section is well described and most details (concentrations, incubation time,…) are provided. Nonetheless, some important information is missing: add in 2.1 sex of animal; add n number for 2.3, 2.4, 2.6, 2.7, and 2.9.

Can the authors comment on the use of different senescent markers for the in vivo and in vitro studies? In vivo, they tested for p16 while γH2AX was used in the primary cardiac fibroblasts experiments.

Since Gata4 is observed in interstitial cells in the αMHCiCre:Gata4fl/fl and Gata4 expression is still observed (Fig 4C), to validate the KO model, cardiomyocytes should be isolated and PCR tested for Gata4 as done previously in the fibroblast KO model.

Minor comments:

Introduction- remove ‘protein’ from first paragraph every time it is mentioned that fibroblasts produce ECM proteins, since ecm is composed of proteins, growth factors, proteoglycans…

There are a few grammatical errors throughout the text, suggest further revision.

There are a few grammatical errors throughout the text, suggest further revision.

Author Response

In this manuscript the authors explore the role of the cardiogenic transcription factor Gata4 in fibroblast senescence in homeostasis. The authors discovered that Gata4 is responsible for cardiac fibroblast senescence; moreover, this was specific to cardiac fibroblasts and did not occur in cardiomyocytes. These findings are significant and present a new avenue of research for the treatment of fibrosis in the heart.

  • The method section is well described and most details (concentrations, incubation time,…) are provided. Nonetheless, some important information is missing: add in 2.1 sex of animal; add n number for 2.3, 2.4, 2.6, 2.7, and 2.9.

Thank you for your valuable suggestion. I've taken your advice into account and included the gender of the animals in section 2.1. You can find this information on page 2, line 80. Moreover, We've indicated the 'n' number for each experiment. For these details, please refer to lines 115, 127, 148, 156, 162, 168, 238, 241, 266, 299, 303, 308, 345, 400, 437, 446, and 449.

  • Can the authors comment on the use of different senescent markers for the in vivo and in vitro studies? In vivo, they tested for p16 while γH2AX was used in the primary cardiac fibroblasts experiments.

Thank you for pointing this out.  We appreciate the reviewer’s careful and rigorous examination of the data. We used both makers for in vivo and in vitro experiments. Please check in vitro p16 staining images in Supplementary Materials Figure S3.

  • Since Gata4 is observed in interstitial cells in the αMHCiCre:Gata4fl/fl and Gata4 expression is still observed (Fig 4C), to validate the KO model, cardiomyocytes should be isolated and PCR tested for Gata4 as done previously in the fibroblast KO model.

I appreciate your insightful suggestion. To show the cells' identities, we have added images of Gata4 and TnI (a cardiomyocyte marker) staining to Supplementary Materials Figure S4. In this figure, the control group showed Gata4 expression in both cardiomyocytes and non-cardiomyocytes. However, in the cardiomyocyte-specific Gata4-/- group, all Gata4 positive cells are found outside the cardiomyocytes, indicating their exclusive expression in interstitial cells.

Minor comments:

Introduction- remove ‘protein’ from first paragraph every time it is mentioned that fibroblasts produce ECM proteins, since ecm is composed of proteins, growth factors, proteoglycans…

We agree with your comment. We have deleted “protein” from the first paragraph. Please see lines 29 to 37 of the revised manuscript.

There are a few grammatical errors throughout the text, suggesting further revision.

Thank you for your comments. We have made multiple changes to correct our grammatical errors. Please see the changes marked in red text.

Reviewer 3 Report

In the submitted manuscript Zhang et al used an inducible model of Gata4 knockout to determine if Gata4 deficiency influenced senescence.

There are major concerns regarding data presented herein.

Morphometric or physiological data was not included.

All figures are not legible. Better quality figures need to be included. Light microscopy images also need to be included of the tissue being shown demonstrating it is heart tissue.

Not once did the authors convincingly provide data that the cells described were truly fibroblasts. Additional studies need to be  performed to confirm fibroblasts are present. Without these studies, rigor and reproducibility are questionable.

Mice section is missing NIH required language regarding animal usage. Justification of age of animals needs to be cited. How was euthanasia affected?

Ensure all company names and catalog numbers are included.

What was the media used for cells in adenovirus infection?

What were the gating parameters for FACS

Primers need to be included for qPCR.

Overall the findings were questionable due to lack of convincing data describing role of Gata4 knockout in fibroblast senescence. 

English is acceptable. 

Author Response

In the submitted manuscript Zhang et al used an inducible model of Gata4 knockout to determine if Gata4 deficiency influenced senescence.

There are major concerns regarding data presented herein.

  • Morphometric or physiological data was not included.

I appreciate your valuable input. We have now incorporated the Echocardiography results in Supplementary Figure S5. In this figure, we present the Ejection Fraction results for Control, Tcf21iCre: Gata4fl/fl, and αMHCiCre:Gata4fl/fl groups. The findings suggest that heart function does not decrease after the deletion of Gata4.

  • All figures are not legible. Better quality figures need to be included. Light microscopy images also need to be included of the tissue being shown demonstrating it is heart tissue.

Thank you for your suggestion. We have included Troponin I (TnI, a cardiomyocyte marker) staining results in the supplementary materials to confirm that the examined tissue is indeed cardiac tissue. Please refer to Supplementary Materials Figure S4 for more details.

  • Not once did the authors convincingly provide data that the cells described were truly fibroblasts. Additional studies need to be performed to confirm fibroblasts are present. Without these studies, rigor and reproducibility are questionable.

Thank you for your suggestion. Tcf21 is a well-known marker for cardiac fibroblasts, and Tcf21iCre mice have been widely used in cardiac fibroblast research [1-2]. In our study, we crossbred Tcf21iCre mice with Rosa26-CAG-loxP-Stop-loxP-tdTomato mice. Consequently, all tdTomato-positive cells identifiable in the heart are Tcf21 positive, indicating they are cardiac fibroblasts. Please refer to Figure 1 for more details.

  1. Acharya, A., et al., Efficient inducible Cre-mediated recombination in Tcf21 cell lineages in the heart and kidney. Genesis, 2011. 49(11): p. 870-7.
  2. Song, K., et al., Heart repair by reprogramming non-myocytes with cardiac transcription factors. Nature, 2012. 485(7400): p. 599-604.
  • Mice section is missing NIH required language regarding animal usage. Justification of age of animals needs to be cited. How was euthanasia affected?

Thank you. We have added this information to the Methods section. Please refer to the details in lines 80-81 and 180-184.

  • Ensure all company names and catalog numbers are included.

Thanks for the suggestion. We have added the company names and catalog numbers for all reagents we used in the experiment. Please refer to the details in lines 140, 142, 151, and 173,

  • What was the media used for cells in adenovirus infection?

Thanks for your question. We followed the University of Iowa Viral Core's Adenovirus infection protocol for adenovirus infection. During the infection process, we use DMEM supplemented with 2% FBS. Please see lines 133-135.

  • What were the gating parameters for FACS

Thank you for your suggestion. FACS results have been added. Please check the Supplementary Materials Figure S2.

  • Primers need to be included for qPCR.

Thank you for your suggestion. qPCR primers information has been added. Please check the Supplementary Materials Table S1.

Round 2

Reviewer 3 Report

The authors have taken steps to answer the reviewers questions. There are remaining concerns. I have highlighted some of these concerns in both the methods and results. Overall, there is a lack of clarity in these areas. 

Institutional review board statement needs to be included in "Mice" section. Additionally, authors need to include statements regarding the following:

1. How were the animals housed and care for

2. How was euthanasia effected, including primary and secondary methods of euthanasia

3. A statement regarding organ excising needs to be included

4. The following statement needs to be included in all animal studies. "All studies conformed with the principles of the National Institutes of Health “Guide for the Care and Use of Laboratory Animals,” (NIH publication No. 85-12, revised 1996), and the protocol was approved by..."

5. What animals were used as controls?

6. Include a table with body weights, heart weights, etc.

7. n values need to be clarified. Are the authors indicating a replicate with the number of animals, number of hearts, number of slides labeled, etc.

Include the ages of the mice used in each of the figures as well as what controls were used. Ages of animals are important to denote, particularly when the methods describe 2 age groups. Differences in sex were not presented. Authors described in methods the use of male and female animals. What is the sex of the animals presented in representative images?

Write out all abbreviations in figure legends.

Multiple scale bars were used in figures. Please identify in figure legends which scale bars pertain to which images. Also, please check to make sure all scale bars are correct, particularly Figure 1. If images were resized please indicate that in the figure legend. Light microscopy images need to be included in Figure 1. In fact all figures should also included light microscopy images so readers can discern image details and ensure authors are providing heart histological sections. Arrows in Figure 1 are not necessary and are distracting.

Indicate which region of the histological images the higher magnification was obtained. Figure 2A does not appear to come from the same histological sections.

Were images taken in Figure 2A and 2B from the same heart or different hearts? If not please indicate. There was on DAPI staining on the p16 Tnl DAPI however staining was shown on p16 DAPI in Figure 2B.

In Figure 3, were fibroblasts cultured? If so, details regarding culturing techniques need to be included. Alpha-smooth muscle actin is a marker for myofibroblasts, and should not be used to identify not stressed fibroblasts. The presence of a-SMA indicates the stained fibroblasts are activated. Additional fibroblast confirmation studies need to be performed as Figure 3A morphologically appears to be a heterogenous culture and not purely composed of fibroblasts. Figure 3B would benefit from DAPI staining.

In Figure 4B, arrows are not necessary to highlight findings. How was echocardiography performed? Were measurements taken from B-mode or M-mode images? Methods need to be updated to reflect echo measurements taken in Supplementary Figure 5. A %Ejection Fraction of 90% is extremely high. Was tamoxifen administered to control animals?

All supplementary figure legends need to be more descriptive and similar to those in the body of the manuscript.

The discussion is acceptable and accurately explains the findings. 

While the authors present evidence regarding fibroblast and Gata4 in senescence, concerns regarding the manuscript need to be addressed before moving forward with publication.

Author Response

Dear Reviewers,

Thank you for your thoughtful suggestions and insights. We have taken them into account and made revisions to our manuscript accordingly. Please find the attached file detailing our modifications in response to your recommendations.

Best Regards,
Zhentao

Round 3

Reviewer 3 Report

The authors have addressed all issues raised in previous reviews. Minor notes regarding: 1) redundant sentence line 79-80 and 2) use of "white arrows" in figure legends was noted.